# High Pressure Deuterium Passivation of Charge Trapping Layer for Nonvolatile Memory Applications

**DOI:** 10.3390/mi12111316

**Published:** 2021-10-27

**Authors:** Jae-Young Sung, Jun-Kyo Jeong, Woon-San Ko, Jun-Ho Byun, Hi-Deok Lee, Ga-Won Lee

**Affiliations:** Department of Electronics Engineering, Chungnam National University, Daejeon 305-764, Korea; sjy5290@o.cnu.ac.kr (J.-Y.S.); jjk1006@cnu.ac.kr (J.-K.J.); kowoon98@cnu.ac.kr (W.-S.K.); spy04043@o.cnu.ac.kr (J.-H.B.); hdlee@cnu.ac.kr (H.-D.L.)

**Keywords:** deuterium high pressure anneal, flash memory, silicon nitride, retention

## Abstract

In this study, the deuterium passivation effect of silicon nitride (Si_3_N_4_) on data retention characteristics is investigated in a Metal-Nitride-Oxide-Silicon (MNOS) memory device. To focus on trap passivation in Si_3_N_4_ as a charge trapping layer, deuterium (D_2_) high pressure annealing (HPA) was applied after Si_3_N_4_ deposition. Flat band voltage shifts (ΔV_FB_) in data retention mode were compared by CV measurement after D_2_ HPA, which shows that the memory window decreases but charge loss in retention mode after program is suppressed. Trap energy distribution based on thermal activated retention model is extracted to compare the trap density of Si_3_N_4_. D_2_ HPA reduces the amount of trap densities in the band gap range of 1.06–1.18 eV. SIMS profiles are used to analyze the D_2_ profile in Si_3_N_4_. The results show that deuterium diffuses into the Si_3_N_4_ and exists up to the Si_3_N_4_-SiO_2_ interface region during post-annealing process, which seems to lower the trap density and improve the memory reliability.

## 1. Introduction

Although research on ReRAM, MRAM, PCM etc. is being actively conducted for next-generation nonvolatile memories [1,2], the demands for Silicon-Oxide-Nitride-Oxide-Silicon (SONOS) flash memory are still dominant. SONOS memory can obtain large capacity easily and have a simple circuit structure. Due to these advantages, the memory has been continuously developed since its invention, improving its storage capacity and reliability [3]. In SONOS, to optimize the silicon nitride (Si_3_N_4_) film which is used as charge trapping layer (CTL) of the device is very important for the performance enhancement. Substantial research has been carried out to improve the Si_3_N_4_ by varying Si/N ratio during deposition, post-annealing condition, or imbedding nano particles [4,5,6,7,8].

Several works on deuterium (D_2_) passivation of Si_3_N_4_ with high pressure annealing (HPA) has been also reported. In the conventional CMOS transistors, D_2_ HPA has been used for the hot carrier reliability [9,10,11]. HPA is known to enhance the deuterium incorporation at the Si/SiO_2_ interface, and the deuterium-passivated silicon materials and interfaces are more electrically stable and reliable. In the case of the memory device, Tanaka’s group at Toshiba Corporation has reported that the MONOS devices with the memory window increment and improve reliability by applying D_2_ treatment in 2002 [12]. With variations in the pressure of D_2_, they show that D_2_ annealing can be used to terminate the traps. In 2004–2005, S.M. Choi’s group checked the high temperature (900 °C) D_2_ annealing effect on the memory device [13,14], and it showed excellent program/erase endurance and a significantly reduced charge loss rate. They focused on D_2_ annealing to passivate the interface traps between the tunneling oxide and Si substrate. The decreased interface trap was confirmed by the quasi-static capacitance measurement. In 2016, L. Breuil’s group investigated D_2_ effect on poly-Si channel with trap passivation at the interface between the ONO layer and the poly-Si channel and in the bulk poly-Si [15]. The D_2_ high pressure annealing has been conducted at the last step of the fabrication process. The results show drive current (I_ON_), subthreshold swing (SS) improvement after D_2_ annealing, which points to a reduction of the defects at the channel and Si-SiO_2_ interface. Threshold voltage (V_TH_) is also reduced due to the passivation of negative fixed charges in the ONO stack. However, in this study, retention characteristics seem not to be affected. In 2020, J.M. Yu’s group at Korea Advanced Institute of Science and Technology investigated the high-pressure D_2_ annealing effects on Gate-All-Around SONOS memory [16]. Maximum transconductance (g_m,max_), SS, and I_ON_ were extracted, which were improved after D_2_ annealing. The low-frequency noise (LFN) measurement was used to extract the oxide trap density (D_OT_) and it was found that the tunneling oxide-channel interface trap decreases by the trap passivation effect of deuterium annealing.

In this study, D_2_ high pressure annealing (HPA) at low temperature is suggested to suppress the shallow trap formation of Si_3_N_4_. Unlike previous research where D_2_ annealing is applied after the metallization and the analyses are concentrated on the oxide-channel interface traps, the suggested process is applied before the metal deposition and the passivation effect of the traps in silicon nitride is focused on.

During Si_3_N_4_ deposition, stoichiometric Si_3_N_4_ is formed and nonstoichiometric bonding would be created. In nonstoichiometric bonding, there may be silicon vacancies and nitrogen vacancies, and some studies found out that the Si-O-N bond caused by oxygen diffusion to nitrogen vacancy makes shallow traps [17,18]. The oxygen ions can migrate from the tunneling oxide. The suggested D_2_ passivation is intended to cure the nitrogen vacancies and suppress the trap generation (similar to the oxygen complex bond). If the deuterium ion occupies the vacancy site, Si-O-N bond cannot be formed even though the oxygen ion migrates. Compared with H_2_ passivation, D_2_ passivation can cure defects more stably because deuterium forms stronger bond than hydrogen [19]. Via passivate shallow traps with D_2_, the memory window can be decreased, but the retention characteristic is expected to improve.

The fabricated devices are programed and then flat band voltage (V_FB_) is extracted in retention mode at a high temperature via CV measurement. Based on the experimental results, trap energy distribution of Si_3_N_4_ is extracted based on the thermal activated retention model. To find out D_2_ profile and bonding formation in Si_3_N_4_, secondary ion mass spectroscopy (SIMS) and Fourier transform infrared spectroscopy (FT-IR) are analyzed. FT-IR spectroscopy is often used to see atomic bonding in materials [20,21] where each atomic bonding absorbs specific wavelength of light. Furthermore, the greatest benefit of FT-IR is its high sensitivity in order to inspect light atoms such as hydrogen and deuterium.

## 2. Experiments

A metal-nitride-oxide-silicon structure capacitor was fabricated and Figure 1 shows the process flows. Contrary to the SONOS structure, the blocking oxide is skipped to remove the process effect by blocking oxide deposition, which is also preferable for the direct correlation of electrical characteristics with SIMS profile whose sample is nitride/oxide/Si substrate. With p-type silicon substrate, SiO_2_ for tunneling oxide was grown to 7 nm via dry oxidation. Then, the Si_3_N_4_ for CTL was deposited via plasma enhanced chemical vapor deposition (PECVD) with a thickness of 15 nm. Next, D_2_ high pressure annealing was performed on some samples. D_2_ 6%/N_2_ 94% forming gas was used and high-pressure annealing condition was 450 °C, 10 atm. Afterward, post-annealing was done in 600 °C in N_2_ ambient. Ti 100 nm was deposited via RF sputter for use as a top and bottom gate. Table 1 shows the experimental conditions. The as-deposited sample functions as a reference, and the D_2_ HPA sample and D_2_ HPA + post-anneal sample are for testing the temperature effect in D_2_ treatment. For the electrical analysis, CV was measured using a Hewlett Packard 4284A precision LCR meter. Programming was completed via same program voltage using Hewlett Packard 41501A pulse generator. The thermal activated retention model is used to extract the trap energy distribution of Si_3_N_4_ based on the measurement. Physical analyses were also used to support the experiment results. D_2_ is well known for possessing a heavier ion than H_2_, but they both are still very light ions compared with other atoms. Thus, there are not many methods to detect D_2_ in the silicon nitride layer. In this study, SIMS and FT-IR were used to find out D_2_ profile and bonding formation in the silicon nitride after the annealing process.

## 3. Results and Discussion

### 3.1. Electrical Analysis

#### 3.1.1. Memory Window

Figure 2a shows the I-V measurement results of the fabricated devices and shows similar breakdown properties of NO stack independent of annealing conditions. Figure 2b is the CV measurement results at the initial and program state. The maximum capacitance is determined by the nitride/oxide thickness and the values are observed to be similar in all samples, which indicates that the NO thickness is not changed by the annealing condition. The programming pulse width was 0.1 s, and the magnitude was 20 V for one pulse time. Sweep mode with long integration time was used in 1 MHz frequency. Compared with the reference, the device with D_2_ HPA shows a decreased memory window and this effect was accelerated in one experiment with additional post-annealing. This seems to be the result of trap curing in Si_3_N_4_ bulk and interface [22].

#### 3.1.2. Retention Characteristics

Retention properties are measured at various temperatures. The flat band voltage (V_FB_) decrement was measured with CV measurement in retention mode after programming. The retention characteristic was quantified with the percentage of V_FB_|_decrease_ by V_FB_|_prgram-initial_. As shown in Figure 3a, compared with the reference (which shows a 27.53% V_FB_ decrement), the device subjected to D_2_ HPA exhibits slight improvement, with a 21.37% decrement. Then, additional post-anneal activity makes a significant difference in V_FB_ decrease during retention mode. Figure 3b is a comparison result of charge loss amount at various temperatures, which shows that D_2_ passivation effects become clearer at higher temperatures. The extracted values are summarized in Table 2.

#### 3.1.3. Trap Energy Level Distribution

Trap energy level of CTL can be extracted by measuring the charge loss through the tunneling oxide. In this experiment, it is assumed that most of the shallow traps are located at or near the interface between Si_3_N_4_ and SiO_2_ tunneling oxide and the amount of charge loss through the metal gate can be ignored. To extract the trap density in Si_3_N_4_, thermal activated retention model is used like follows [23].
(1)$$\frac{\partial \Delta V_{TH}}{\partial \log\left(t\right)}=-2.3k_{B}TX_{N}\left(\frac{X_{N}}{2\varepsilon_{N}}+\frac{X_{OX}}{\varepsilon_{OX}}\right)g\left(E_{TA}\right)$$
(2)$$E_{TA}=k_{B}Tln\left(AT^{2}t\right)$$
(3)$$A\;=2\sigma_{n}\sqrt{\frac{3k_{B}}{m^{*}}}{\left[\frac{2\pi m^{*}k_{B}}{h^{2}}\right]}^{3/2}$$

Equation (1) shows the charge decay model in relation to V_TH_ shift according to time at specific temperature. E_TA_ is the energy level expressed like Equation (2). In Equation (3), σ_n_ is the capture cross-section and m* is the effective mass of electron in the silicon nitride. Measurements were conducted at 85 °C.

Figure 4a shows the extracted energy distribution of trap in each process condition based on the experimental results of charge loss according to the retention time as shown in Figure 4b. The results show that D_2_ passivation reduces the amount of trap densities in the band gap range of 1.06 eV~1.18 eV, and this suppression seems to be reinforced with post-annealing. This result explains the program window reduction in D_2_ HPA, as mentioned before.

### 3.2. Physical Analysis

#### 3.2.1. Secondary Ion Mass Spectroscopy

SIMS can be used to find out the D_2_ profile in Si_3_N_4_ [24,25]. The selected elements Si, O, N, D, and depth profiles were obtained with a commercial SIMS instrument (CAMECA IMS 7f). The primary ions were 6 keV Cs^+^ at 10 nA and raster into a 250 μm × 250 μm with a detected area of 63 μm in diameter. Figure 5 shows the SIMS profile results for as-deposited silicon nitride and D_2_ passivated Si_3_N_4_, including the post-anneal process. As-deposited Si_3_N_4_ has a negligible quantity of deuterium, but in the sample treated with D_2_ high pressure annealing, it can be seen that deuterium exists from the surface side. Furthermore, with additional post-annealing, it was confirmed that deuterium diffuses into the Si_3_N_4_ and exists up to the Si_3_N_4_-SiO_2_ tunneling oxide interface region. It is possible that deuterium can be dissociated during the post-annealing period, but the SIMS results indicate that deuterium affects silicon nitride property during N_2_ 600 °C anneal. That is, even if some injected deuterium could dissociate and diffuse out to air at 600 °C, some could diffuse into the layer and form bondage with Si and N atoms. Furthermore, previous research had conducted D_2_ high pressure annealing even at 900 °C [12]. This result supports our theory as why V_FB_ in the initial state is shifted to the left and the memory window decreases.

#### 3.2.2. Fourier Transform InfraRed Spectroscopy

To analyze the change in atomic bonding in Si_3_N_4_, it is needed to detect the deuterium bonding. In FT-IR, each bonding absorbs specific wavelength of Infrared light and the results don’t affect each other. The analysis can be employed for detecting and determining bond densities of light atoms such as H_2_ or D_2_. In 2008, G. Scardera investigated high temperature annealing effect on silicon-rich silicon nitride films using FT-IR spectroscopy [20]. Si-H bonding is detected by FT-IR measurement, allowing for investigation of changes in bonding ratio after annealing process. In 1995, Z. Lu’s group also used FT-IR spectroscopy to check the RTA effect on a-Si:N:H(D) films, where Si-H and Si-N-H bonds are detected and Si-D and Si-N-D peaks are also detected [21].

In this experiment, FT-IR spectroscopy is applied to find out the D_2_ passivation effect on silicon nitride bonding structure. Figure 6 shows the results of FT-IR spectroscopy on Si_3_N_4_ with D_2_ passivation. A Thermo-Nicolet 5700 FT-IR spectrometer was used. With reference to previous studies, the range of the SiN-D peak is around 2400 cm^−1^. As shown in Figure 6, the sample with D_2_ HPA with 600 °C annealing shows slightly increased absorbance in the rage of 2375 cm^−1^. Considering the difficulty of detecting the light atoms (such as H_2_ or D_2_), more precise physical method should be studied.

## 4. Conclusions

In this study, the effects of D_2_ HPA on Si_3_N_4_ are investigated in MNOS-type flash memory device. To focus on silicon nitride’s trap control, D_2_ passivation is conducted directly on Si_3_N_4_ films before metal deposition. The results show the memory window decreased after D_2_ passivation, but the charge loss in retention mode after the program is suppressed, which becomes clearer as temperature increases. The D_2_ passivation effect seems to be reinforced with post-annealing. Trap energy distribution based on the thermal activated retention model is also extracted to compare the trap density. The results show that D_2_ passivation reduces the amount of trap densities in the band gap range of 1.06 eV–1.18 eV. SIMS and FT-IR spectroscopy are also applied to find out the deuterium profile and bond structure in Si_3_N_4_. SIMS results show that deuterium diffuses into the Si_3_N_4_ and exists up to the Si_3_N_4_-SiO_2_ tunneling oxide interface region, which demonstrates the possibility of the deuterium passivation of shallow traps near Si_3_N_4_ and SiO_2_ interface.

## Figures and Tables

**Figure 1 micromachines-12-01316-f001:**
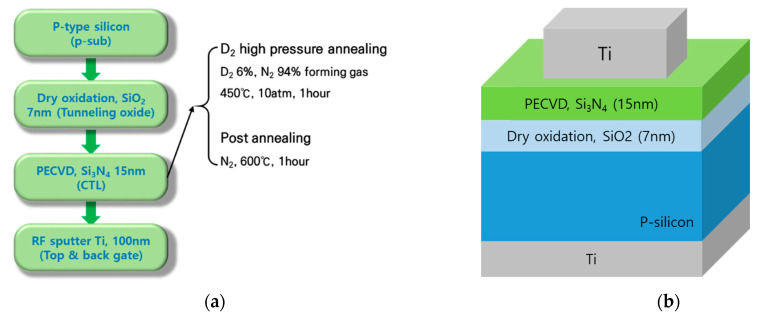
(**a**) Fabrication process and (**b**) the cross view of the capacitor-type device with metal nitride oxide silicon (MNOS) structure.

**Figure 2 micromachines-12-01316-f002:**
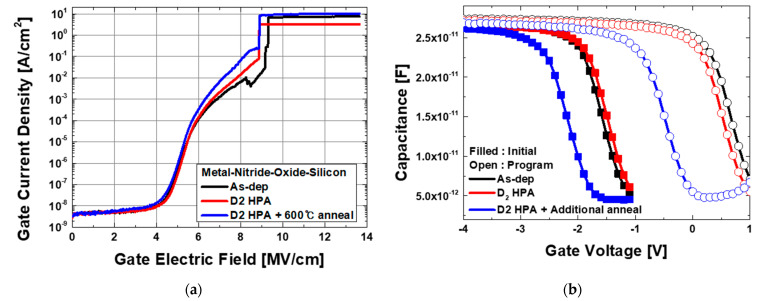
(**a**) The I-V measurement results of the fabricated devices and (**b**) the CV measurement results at the initial and programmed state. Here, as-dep is a reference device, and D_2_ HPA is D_2_ high pressure annealed device at 450 °C and D_2_ HPA + Post-anneal is D_2_ high pressure annealed device with post-anneal at 600 °C.

**Figure 3 micromachines-12-01316-f003:**
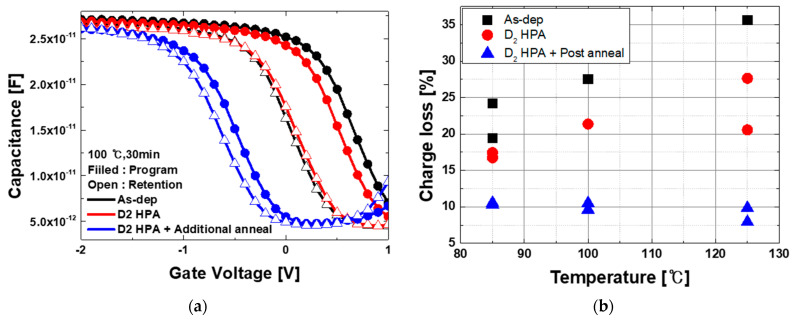
CV measurement results in data retention mode at 30 min after programming. (**a**) CV shift at 100 °C retention mode; (**b**) extraction results of charge loss amount (percent) in retention mode at various temperature. Here, charge loss was calculated by the amount of V_FB_ shift after retention time divided by the memory window.

**Figure 4 micromachines-12-01316-f004:**
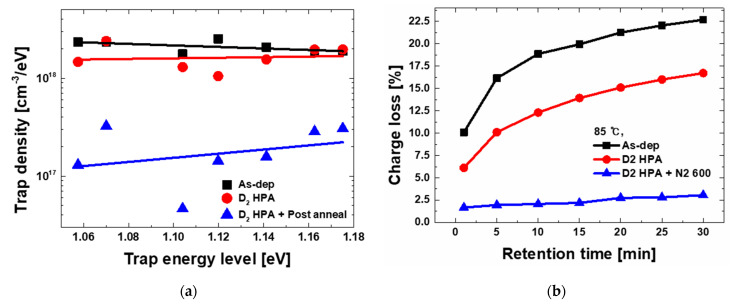
(**a**) Trap energy level distribution is Si_3_N_4_ according to the post treatment. (**b**) Measurement results of charge loss according to the retention time at 85 °C. The extraction is based on thermal activated retention model.

**Figure 5 micromachines-12-01316-f005:**
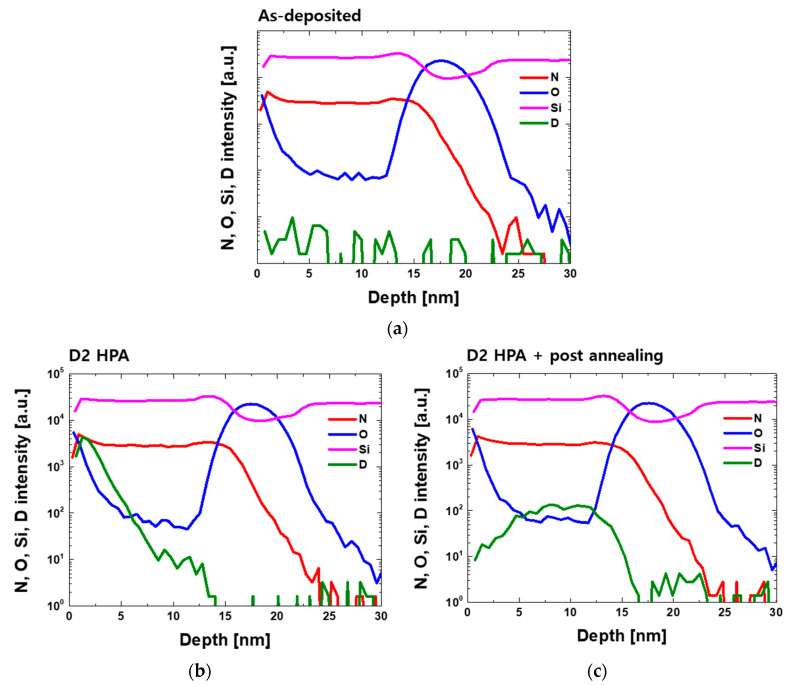
SIMS profile of N, O, Si, D in (**a**) as-deposited silicon nitride; (**b**) D_2_ high pressure annealed silicon nitride; (**c**) D_2_ high pressure anneal with post-annealing silicon nitride.

**Figure 6 micromachines-12-01316-f006:**
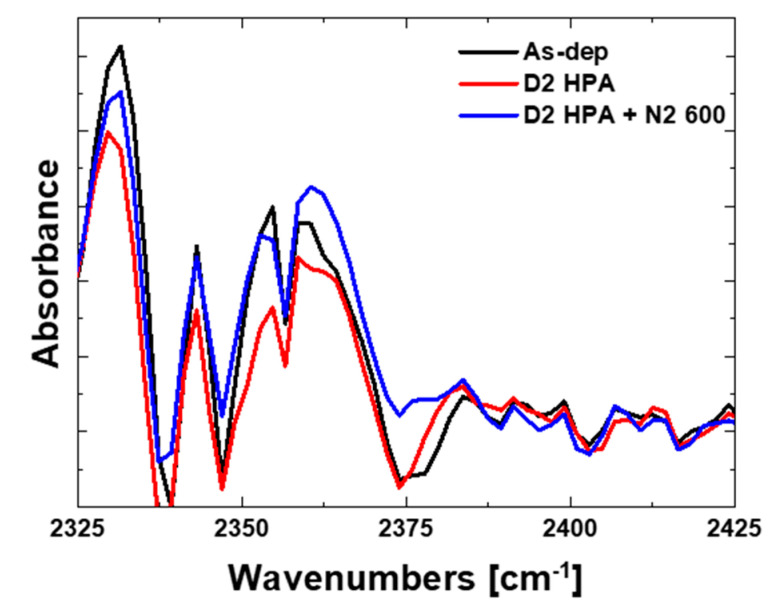
FT-IR result for silicon nitride comparing with the as-deposited sample and the D_2_ passivated sample. The sample with D_2_ HPA with 600 °C annealing shows slightly increased absorbance in the rage of 2375 cm^−1^.

**Table 1 micromachines-12-01316-t001:** Experimental conditions of the post treatment on the silicon nitride, which is performed before the metal deposition.

	As-Deposited	D_2_ HPA	D_2_ HPA + Post Anneal
D_2_, 450 °C, 10 atm	×	O	O
N_2_, 600 °C	×	×	O

**Table 2 micromachines-12-01316-t002:** Charge loss percent of each device in retention mode after programming at various temperature.

	$\mathrm{Charge}\;\mathrm{Loss}\;(\%)=\frac{V_{FB|program}-V_{FB|retention}}{V_{FB|program}-V_{FB|initial}}\times 100$		
Temperature	85 °C	100 °C	125 °C
As-dep	19.43	27.53	35.65
D_2_ HPA	16.75	21.37	27.65
D_2_ HPA + Post-anneal	10.30	10.52	9.84

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
