# Peer review of "High Pressure Deuterium Passivation of Charge Trapping Layer for Nonvolatile Memory Applications"

_micromachines, 2021, doi:10.3390/mi12111316_

Round 1

Reviewer 1 Report

This paper demonstrates the impact of deuterium annealing with respect to Si3N4 passivation. Especially, the interesting points are, 1) they investigated about the annealing effect which performed right after the Si3N4 deposition, and 2) quantitative analyses of bulk traps which are not located at SiO2/Si interface. Hence, I would like to accept this paper if several modifications are clearly included in their revised manuscript.   

1) It would be informative for readers if the authors could add recently published references and related sentences in terms of device reliability in logic devices, in their revised introduction. For example,

[1] Relevance of fin dimensions and high-pressure anneals on hot-carrier degradation, IRPS 2020.

[2] Impact of Post-Metal Annealing With Deuterium or Nitrogen for Curing a Gate Dielectric Using Joule Heat Driven by Punch-Through Current, IEEE EDL 2021.

2) Why the authors did not apply ONO stack for their capacitor? It would be appreciated if brief explanations about the reason why MNOS structure had been applied instead of ONO, could be added on their revised experiments section.

3) Detailed measurement setup for programming condition e.g., pulse width (duration), pulse magnitude, number of pulses should be included for reader’s understandings. In addition, measurement methodologies which have been used for Figs. 2 and 3a, (e.g., double sweep, integration time etc.) should be addressed in detail.

3) Why the figures in Figs. 2, 3a, and 4a are shown with arbitrary unit? Unless there are specific reasons, the authors should disclose the values for reader’s understandings.

4) The authors claimed that deuterium can be diffused and bonded with Si or N after the PMA of 600C. However, at the temperature higher than 450C deuterium dissociation can be of concern. Please, comment about this concern and added related references in their revised manuscript.

Minor

Please check page 2 and 3. Several words ‘D2’ are colored in red.

Reviewer 2 Report

The authors of the manuscript “High-Pressure Deuterium Passivation of Charge Trapping Layer for Nonvolatile Memory Applications” present a study on the influence of deuterium annealing on the trapping properties of MNOS type memory stacks. Although the work is original and quite interesting the manuscript suffers from problems.

  • The manuscript is quite hard to read. The authors must use some kind of editing program to improve the text and thus make it more readable.

1) The authors do not comment on the C-V characteristics of the structures. How do the different processes employed influence the electrical characteristics of the structures as determined by the C-V characteristics? This is evident for example in Fig.2 where the capacitance of the D2 HPA + Post Anneal sample shows a response at inversion. In addition, normalization of the capacitance has been performed. Why? Do the processes alter the dielectric stack properties as well? This is significant since, from an analysis of the C-V characteristics, important points on the dielectric behavior of the stacks can be drawn and more importantly on the influence of the process on them. 

2) The second point is the absence of I-V characterization. Since we are talking about MNOS structures, the I-V characteristics reveal the high field leakage properties of the structures under study. How do the various processes influence the leakage properties of the stack? In this work, this issue is not addressed.

3) The third point and the most important, is the use of the so-called “thermal activated retention model” by Yang (Larr) Yang and Marvin H. White (reference [20] in which the first author is missing!) states clearly that this model is applied in the presence of a blocking oxide, which in our case is absent. So the authors assume that all of the charges are lost to the side of the tunnel oxide, which may be not the case. Consequently, Figure 4 and the conclusions drawn from it can be considered invalid.

4) Another point of concern is the FTIR measurement. The authors use previous works which refer to a-Si:N:H (D) samples on which they conclude that the Si-N-D bond gives a peak at around 2400 cm-1. Based on their graph (Figure 6) all samples exhibit 4 peaks within the wavenumber range 2325-2375 cm-1, and also very close to the noise of the spectra. Which one is the Si-N-D? This point is unclear.

On the positive side, the authors have two important experimental facts.

1) the low charge loss in retention mode.

2) the SIMS deuterium profile which indicates that the deuterium is distributed within the silicon nitride trapping layer.  

Reviewer 3 Report

The work titled “High Pressure Deuterium Passivation of Charge Trapping Layer for Nonvolatile Memory Applications” by Sung et al. introduce the high pressure deuterium (HPD) passivation effect into MNOS memory device. Deuterium HPA + post anneal process lower the trap density and therefore make the memory reliability improved. In addition, the author uses SIMS profile and FT-IR spectroscopy to analyze deuterium HPA impact for Si3N4 interface. This work is interesting and timely. However, the following questions need to be addressed carefully. 
1. The authors indicate that memory retention improvement is an important advantage of HPD passivation, but only one retention time test result shown. More retention results are suggested to present to enrich the data.
2. In figure 2 and figure 3, what are the electrical test parameters? Such as the programming pulse parameters. The authors should point out that in the manuscript.
3. In figure 6, as-deposited, D2 HPA and D2 HPA + post annealing data lines look almost identical. It’s hard to recognize the SiN-D peak difference between the three experimental conditions. The author should add some figure signs or enlarged figure inset to clearly clarify the change of Si-O-N bond density.
4. The title mentions nonvolatile memory application, but no direct data is depicted in the article. The reviewer suggests showing some statistical data of Vth before and after programming.
5. In line 197, the word “boding” should be corrected to “bonding”. Please carefully check other typos.

Round 2

Reviewer 1 Report

The author's responses are clear, and the quality of revised manuscript is highly improved.  Hence, I would like to recommend to publish this paper in present form. 

Reviewer 2 Report

In the revised version the manuscript under consideration has been improved. 

I had detected some minor typos and a sentence that requires improvement.

page 1 line 12 change date to data

page 2 line 89 correct plasm to plasma

page 3 lines 100 to 101, please improve the sentence 

Reviewer 3 Report

The authors have addressed all my questions and could be accepted now. Note that the authors should be rigorous, it seems that wrong content (emails related to other papers) is included in the response letter.